# OpenReview forum: "A Positive Case for Faithfulness: Explanations Help Predict Model Behavior"
_ICML.cc/2026/Conference — ICML 2026 regular_

### Official Review · Reviewer_V3kx · 2026-03-11

**Soundness:** 2
**Presentation:** 2
**Significance:** 2
**Originality:** 1
**Overall Recommendation:** 3
**Confidence:** 3

**Summary:**

This paper introduces Normalized Simulatability Gain (NSG), a faithfulness metric addressing the "vanishing signal" problem in existing LLM explanation methods that rely on detecting adversarial vulnerabilities. NSG measures whether an explanation helps predict a model's behavior on related counterfactual inputs, rather than just checking for logical consistency. Evaluating 18 frontier models across health, business, and ethics domains, the authors find that self-explanations consistently encode valuable predictive information. They demonstrate that self-explanactions contain privileged internal knowledge inaccessible to external observers.

**Compliance With Llm Reviewing Policy:**

Affirmed.

**Final Justification:**

I acknowledge the points made by authors and agree to raise my score.

**Key Questions For Authors:**

N/A

**Strengths And Weaknesses:**

Although the paper has conducted substantial experiments on many current LLMs, I don't find it very informative or valuable. The main problem is the core concept itself—the faithfulness of explanations is a well-studied topic in XAI with well-understood definitions and properties.

In one sentence, the paper argues that when a base model explains its predictions, those explanations can help other models better predict the base model's behaviour. Sure, I agree with the authors that this is true. But isn't that trivial?

If the goal is to study the quality of LLM explanations, I would suggest the authors consider existing explanation evaluation metrics such as those presented in [1].

As it stands, I feel the authors just reinvented a known XAI metric: Counterfactual Simulatability, which has been studied in a LLM context [2],[3], and conducted loads of experiments.

I don't think this is strong enough for an ICML paper.

In case the links don't work, the paper I mentioned are titled: From Anecdotal Evidence to Quantitative Evaluation Methods: A Systematic Review on Evaluating Explainable AI, Do Models Explain Themselves? Counterfactual Simulatability of Natural Language Explanations, and Counterfactual Simulatability of LLM Explanations for Generation Tasks.

[1]: https://arxiv.org/pdf/2201.08164 (From Anecdotal Evidence to Quantitative Evaluation Methods: A Systematic Review on Evaluating Explainable AI)
[2]: https://proceedings.mlr.press/v235/chen24bl.html (Do Models Explain Themselves? Counterfactual Simulatability of Natural Language Explanations)
[3]: https://aclanthology.org/2025.inlg-main.38/ (Counterfactual Simulatability of LLM Explanations for Generation Tasks)

---

> ### Author Rebuttal · Authors · 2026-03-30
>
> Thank you for your review. We’ll address each of your comments in turn.
>
> ---
>
> ### **Explanation faithfulness**
> We respectfully disagree with the reviewer’s claim that the concept of explanation faithfulness has well-understood definitions. For instance, **How should we define and evaluate faithfulness? [1]** argues that faithfulness is “lacking a standard definition”, and **Towards Faithful Model Explanation in NLP [2]** states faithfulness “does not have a universally accepted technical definition”. While the *intuition* is agreed upon, operationalizing faithfulness remains an open question to which we contribute.
>
> We also respectfully disagree that explanations being useful for predicting model behavior is trivial. Prior work **[2-8]** argues unequivocally that many self-explanations do *not* help prediction. In fact, Chen et al. (2023) **[7]** conclude that “State-of-the-art LLMs generate misleading explanations that lead to wrong mental models”. Our work also shows concrete examples where explanation can mislead prediction (Figure 6).
>
> ### **Relationship to counterfactual simulatability**
> Our work builds on counterfactual simulatability as studied by Chen/Limpijankit **[7/8]**. Our contribution is not introducing a separate paradigm, but to make a key conceptual improvement. Counterfactual simulatability measures predictor accuracy without reference to any baseline, so it is unclear whether accuracy is due to the explanation quality or the predictor’s baseline prediction rate. We address this by comparing predictor accuracy with and without explanations, isolating the explanation's effect on simulatability. The core claims of our paper could not be made using only existing counterfactual simulatability frameworks. By calling our metric “simulatability gain”, we deliberately signal the relationship to counterfactual simulatability, while making the distinction explicit. We will expand the “Counterfactual simulatability” section to clarify this.
>
> We also note several experimental advantages over Chen/Limpijankit, detailed in the “Originality” response below.
>
> ### **Existing explanation literature citation**
> We thank the reviewer for pointing out **[9]**. We agree that this is a valuable survey of evaluation practices in XAI. The authors introduce 12 desirable properties of explanations, one synonymous with faithfulness:
>
> > “Our property ‘Correctness’ addresses the faithfulness of an explanation” (2.3)
>
> They reference six ways of operationalizing this property (Table 3), together with the explanation types each is suitable for evaluating (feature importance, heatmaps, localization, etc). This table explicitly shows that none of these metrics can apply to textual explanations (p16). The authors call for new evaluation methods to address such gaps (p28), and our work answers this call.
>
> ### **Originality**
> One of the reviewer’s main arguments regards originality. We view originality as one of the core strengths of our work. All of the following are novel contributions:
> - Novel dataset construction using data-driven counterfactuals and allowing multiple factors to vary simultaneously.
> - The definition of NSG as a way to operationalize *faithfulness* (building on counterfactual simulatability).
> - Experimentation on frontier proprietary and open-weight models, showing consistent average faithfulness across models, departing from the existing narrative in the literature (see references under Explanation faithfulness).
> - Feature-level analysis of unfaithfulness.
> - Detection of a statistically significant self-knowledge effect, where a model’s own explanation results in higher NSG.
>
> ### **Presentation**
> We also see presentation as a strength of our paper. The other three reviewers scored this manuscript *good*, *excellent*, *excellent* for presentation. If you have any actionable feedback on the presentation, we would find this valuable.
>
> ---
>
> Thank you again for your review. We hope our clarifications and planned additions have positively impacted your impression, and you will consider increasing your score.
>
> **References (arXiv ID)**
> 1. Towards Faithfully Interpretable NLP Systems: How should we define and evaluate faithfulness? (2004.03685)
> 2. Towards Faithful Model Explanation in NLP: A Survey. (2209.11326)
> 3. Language Models Don't Always Say What They Think: Unfaithful Explanations in Chain-of-Thought Prompting. (2305.04388)
> 4. Reasoning Models Don't Always Say What They Think. (2505.05410)
> 5. Chain-of-Thought Reasoning In The Wild is not Always Faithful. (2503.08679)
> 6. Chain-of-Thought is Not Explainability (https://www.alphaxiv.org/overview/2025.02v3)
> 7. Do Models Explain Themselves? Counterfactual Simulatability of Natural Language Explanations. (2307.08678)
> 8. Counterfactual Simulatability of LLM Explanations for Generation Tasks. (2505.21740)
> 9. From Anecdotal Evidence to Quantitative Evaluation Methods: A Systematic Review on Evaluating Explainable AI. (2201.08164)

---

> > ### Author Rebuttal · Reviewer_V3kx · 2026-04-01
> >
> > Thank you for the detailed rebuttal and clarifications. I appreciate the effort to better position the work and to engage with prior literature. However, after considering your response, I maintain my original assessment and score. My concerns are not about experimental scale or presentation, but about conceptual positioning and novelty relative to the established XAI literature.
> >
> > While we see that faithfulness does not have a single universally accepted formal definition, I do not agree with the characterization that it remains an underdefined or insufficiently studied concept. The XAI community has developed a rich and structured understanding of explanation quality, including faithfulness, completeness, simulatability, sufficiency and necessity, and robustness. In particular, simulatability and counterfactual reasoning have long been recognized as central tools for evaluating explanations. From this perspective, the problem the paper addresses is not new, even if different subfields, e.g., LLMs, emphasize different operationalizations.
> >
> > Regarding the relationship to counterfactual simulatability, I appreciate the clarification that NSG is intended as a refinement rather than a new paradigm. This also reinforces my concern. The proposed metric can be understood as a baseline-adjusted version of existing simulatability measures, where the marginal contribution of explanations is isolated by comparing performance with and without explanations. This, in my view, is an incremental methodological improvement rather than a conceptual advance.
> >
> > I also acknowledge that the paper includes thoughtful design choices, such as the breadth of experiments across multiple models and domains. These bring some empirical value. However, taken together, I still see the contribution as primarily empirical and incremental. The work does not sufficiently engage with or build upon the broader XAI literature on explanation evaluation and instead frames the problem largely within the LLM evaluation context. This leads to a positioning that, in my view, understates how extensively these questions have already been explored in XAI. On the question of whether the findings are trivial, I do not find the central result—that explanations improve predictive simulatability on average—to be surprising given prior work on simulatability and explanation utility.
> >
> > In summary, I view this paper as a rich empirical study with an incremental metric refinement. From my perspective as an XAI researcher, the core ideas and evaluation principles have been well studied, and the paper does not sufficiently acknowledge or build upon that foundation. For these reasons, I do not find the level of novelty sufficient and therefore maintain my original score.

---

> > > ### Author Response · Authors · 2026-04-03
> > >
> > > Thank you for your reply. We’ve organized our response into two parts. *PART 1* clarifies the conceptual contribution in response to the reviewer. *PART 2* discusses the other contributions of this paper. Irrespective of our disagreements, we feel that these other contributions are of value and warrant publication.
> > >
> > > ## PART 1: Continued discussion and response
> > >
> > > ### **Conceptual contributions.**
> > >
> > > Providing a novel operationalisation of an existing concept (faithfulness), in a domain where existing operationalisations are not applicable [9, 10] is, in our view, an important conceptual contribution. The problem of faithfulness is not new, but self-explanation faithfulness cannot be studied without a reliable operationalisation. This sentiment is explicitly noted in the survey the reviewer recommends [9]:
> > >
> > > > “We also see a research opportunity to develop new evaluation methods for Co-12 properties that are currently insufficiently addressed, and to develop variants on existing evaluation methods to make them suited for different types of data and explanations.”
> > >
> > > ### **Building on the XAI literature.**
> > >
> > > As a team of XAI researchers ourselves, this was the original motivation of this project. When we introduce our metric we explicitly say “This approach mirrors the explainable AI literature” in reference to LIME. While we respectfully disagree that the existing current manuscript does not acknowledge XAI sufficiently, we are very familiar with the history of different approaches in XAI and would be happy to further discuss this in the paper.
> > >
> > > ### **Contributions to the self-explanation literature.**
> > >
> > > The goal of the paper is to study LLM self-explanations because these are uniquely high-risk. With LLM-based tools being widely adopted, the extent to which their explanations can be trusted remains generally unclear [6]. Additionally, self-explanations are often viewed as a promising opportunity for model monitorability in AI safety [11]. The dominant narrative in the self-explanations literature is that these explanations are pure post-hoc rationalisations [2,3,4,5,6,7,8,10].
> > >
> > > It is not clear why positive simulatability/utility results for other XAI methods should have any bearing over how surprising this result is because we are evaluating a different explanation method.
> > >
> > > The most important contribution of our paper is to provide evidence contrary to the existing narrative. If you believe our results are unsurprising, then you must also appreciate the importance of shifting the existing narrative and thus the importance of the contribution here.
> > >
> > > ## PART 2: Other contributions and valuable impact
> > >
> > > Our manuscript makes several other meaningful contributions, distinct from the methodological contribution that has been the focus of this discussion. These are highlighted in *Originality* section or our initial response to your review. We want to make sure these are not ignored. Despite our disagreements, we hope the reviewer acknowledges the value and potential impact of the other contributions of this paper and the “rich empirical study” of this manuscript.
> > >
> > > ### Concluding remarks
> > >
> > > A score of 2 is described as containing *technical flaws, weak evaluation, inadequate reproducibility,..., or incomprehensible writing*. We hope the reviewer agrees that none of these characteristics apply to our paper, and they will consider revising their score. We would also welcome any actionable feedback from the reviewer.
> > >
> > > ---
> > >
> > > [1-9] See reference in original rebuttal.
> > >
> > > [10]  “Unfortunately, we do not currently have viable dedicated evaluations for reasoning faithfulness.” (Claude Sonnet 4.5 System Card, Anthropic.)
> > >
> > > [11] Chain of Thought Monitorability: A New and Fragile Opportunity for AI Safety

---

### Official Review · Reviewer_bXva · 2026-03-11

**Soundness:** 3
**Presentation:** 4
**Significance:** 3
**Originality:** 3
**Overall Recommendation:** 5
**Confidence:** 2

**Summary:**

The paper looks at LLM self-explanations, where an LLM justifies or explains its own reasoning behind a decision. They introduce a Normalized Simulatability Gain (NSG) faithfulness metric that measures the improvement in decision making (via accuracy) that is provided by an explanation. Through experiments and measures of NSG, the paper shows that self-explanations are able to improve predictions, making a positive case for self-explanations faithfulness. The experiments are extensive, covering 18 reference models and seven datasets. They also test whether faithfulness scales with model size, but find mixed trends.

**Compliance With Llm Reviewing Policy:**

Affirmed.

**Final Justification:**

My final recommendation is accept. I had some reservations about the motivation and soundness of the paper as I have doubts about LLM 'self explanation', but the authors responded to this in their rebuttal in a way I'm happy with. I though the presentation and clarity was great and I think it is original and significant. That being said, I’m not familiar enough with the related work on LLM explanations specifically to be able to be fully confident about this assessment.

**Key Questions For Authors:**

N/A

**Limitations:**

Yes

**Strengths And Weaknesses:**

The paper is presented well and written clearly, the narrative was easy to follow despite not having full
familiarity with the topic of LLM explanations. The paper is sound, however I have concerns generally about what self explanations represent, whether it’s a genuine explanation or just a plausible narrative created by the LLM, but that’s not a specific critique of this paper but of ‘self-explanations’ more generally. The author(s) offer extensive experimental results to support their findings. The way the author(s) have positioned the paper demonstrates its originality and significance to the reader. That being said I’m not familiar enough with the related work on LLM explanations to be able to confidently comment on the originality and significance, which is why I have given a score of 3 (good) rather than 4 (excellent).

---

> ### Author Rebuttal · Authors · 2026-03-30
>
> Thank you for your thoughtful and positive review!
>
> Your concern is the central motivation for this work: are self-explanations genuine explanations or just plausible narratives? Our goal with NSG is to provide a way to empirically test whether explanations contain predictive information. In this sense, we see our work as contributing a concrete tool for studying precisely the concern you raise.
>
> We appreciate your positive comments about our presentation and the scope of our experiments.

---

> > ### Author Rebuttal · Reviewer_bXva · 2026-04-01
> >
> > I'm still slightly on the fence about self-explanation generally but I don't believe that should hold back the paper. I still recommend accept.

---

### Official Review · Reviewer_uVtH · 2026-03-13

**Soundness:** 3
**Presentation:** 4
**Significance:** 4
**Originality:** 3
**Overall Recommendation:** 5
**Confidence:** 4

**Summary:**

This paper introduces a new metric for measuring the faithfulness of LLM explanations by the name of Normalized Simulatability Metric (NSG), which measures how much an explanation helps an observer predict an LLM's behavior on related inputs. Extensive evaluations reveal that self-explanations improves prediction of model behavior (11-37% NSG), along with the finding that self-explanation improves NSG over an external explanation from a stronger model, which suggests self-knowledge advantage. NSG also reveals that 5-15% of self-explanation can actually be be egregiously misleading, which provides a measured view of the current state of explanations.

**Compliance With Llm Reviewing Policy:**

Affirmed.

**Final Justification:**

This paper introduces a well-motivated and scalable faithfulness metric that addresses a genuine evaluation gap for frontier LLMs. The experimental design is thorough, the results are robust across extensive ablations, and the rebuttal adequately addressed my concerns. I maintain my recommendation of accept.

**Key Questions For Authors:**

1. Though the paper points out how faithful explanations and support AI safety, NSG is lowest on the Moral Machines domain (arguably the domain most closely align with the safety motivation), with several models showing statistically significant negative NSG. This is an interesting result given that we'd like to understand model decision making a priori in these domains but cannot trust their explanation. Beyond the dimension-level RR already in the appendix, can you compute dimension-specific NSG to show which moral dimensions contribute most to the low aggregate faithfulness?
2. NSG is affected by explanation quality, counterfactual set design, and predictor capability. If stronger predictor models were used, would you expect NSG to increase (because they extract more from explanation) or decrease (because the baseline improves and the denominator goes down)? This has implication on whether NSG may also suffer from the vanishing signal problem.
3. Table 2 shows that self-knowledge uplift is consistently positive but modest. Are there any observable trends regarding domains or model properties (e.g. model size, dataset complexity, gap between self and cross-model baseline accuracy) where self-knowledge helps more than others?

**Limitations:**

Yes

**Strengths And Weaknesses:**

Strengths
1. Very strong experimental design with extensive and thorough evaluations and ablations. Evaluation of 18 models with 5 predictor models across model families, closed/open-source, and diverse domains supports generalizable findings.
2. NSG is a well-motivated metric that combines inspiration from key papers from explainable AI on explainability as prediction, alongside other ideas such as data-counterfactual simulatability, normalization for ceiling effects, and using real-world data distributions for counterfactuals.
3. Measuring faithfulness of explanations around what information they reveal rather than what failures they expose reduces the susceptibility to the vanishing signal problem and increases the longevity of this work.
4. The paper presents a measured view of explanations by not only exploring how they can be faithful, but how they can also be egregiously unfaithful at times.

Weaknesses
1. The paper claims in Section 4.3 that the gap between predictor accuracy with explanations and the oracle ceiling represents "significant unfaithfulness despite consistent reference model behavior." However, this conflates explanation unfaithfulness with predictor limitations. A perfectly faithful explanation could still fail to help if a predictor misunderstands the explanation, struggles to perform the necessary logical inference, or suffers from biases that override the explanation's content. The gap is an upper bound on the unfaithfulness rather than a direct measurement of it, and this distinction should be made explicit.
2. NSG is jointly determined by explanation quality, counterfactual set design, and predictor capability. The authors thoughtfully consider their counterfactual set (Section 2.3) and show rank agreement across predictors (Kendall's W = 0.75). However, the dependence on predictor strength is underexplored. Predictor capability affects both the baseline accuracy and the ability to extract information from explanations, likely influencing NSG in complex ways. A more thorough analysis of how predictor strength affects NSG would strengthen the paper.
3. The paper's title and motivating framing make general claims about LLM self-explanations and AI safety, but the evidence is primarily drawn from tabular binary classification tasks with structured features. The authors acknowledge that extension to free-text generation is an open problem (Section 7), and that the relationship between faithfulness in benign versus adversarial settings is unresolved. However, the paper's framing does not adequately reflect the severity of these limitations. The BBQ extension (Appendix A.11) involves only two models, one with NSG statistically indistinguishable from zero. I would suggest tightening the claims to better match the scope of the evidence — for instance, specifying that the positive case for faithfulness applies to structured classification domains in benign settings, rather than implying generality across LLM reasoning.
4. In Section 4.3, features such as Education level and Occupation are described as "technical concepts" in contrast to protected characteristics like Race and Sex. This labeling is misleading — education level and occupation are straightforward socioeconomic categories, not technical in any meaningful sense. The actual distinction being drawn is between protected and non-protected attributes. Relabeling this would clarify an otherwise interesting finding: that unfaithfulness appears driven by difficulty articulating complex feature interactions rather than by training incentives to conceal reliance on sensitive attributes.

Weaknesses 1, 2, and 4 are addressable during rebuttal through clarifications and additional analysis on existing data; weakness 3 is partially addressable through tightened language but the domain scope limitation would require new experiments

---

> ### Author Rebuttal · Authors · 2026-03-30
>
> Thank you for your thorough and positive review! We appreciate the recommendation of acceptance. Your feedback is helpful, and we will edit the manuscript in the following ways.
>
> ---
>
> ### **Weakness 1: Oracle predictors**
> We agree that the distinction between unfaithfulness and predictor limitations is important, and we will delineate this in the camera-ready version (see weakness 2). We will also clarify the definition of an “oracle predictor”. Here, we use the reference model’s modal output as the prediction (the oracle), allowing us to calculate the maximum performance given inconsistency.
>
> ### **Weakness 2/Question 2: Predictor strength exploration**
> Building on the discussion above, we further examine how predictor strength affects NSG components:
>
> | Predictor | Acc (w/ expl) | Acc (w/o expl) | Abs. Gain | False-Change rate (w/o expl) |
> |---|---|---|---|---|
> | Qwen3 32B | 77.3% [77.0, 77.5] | 65.1% [64.9, 65.4] | +12.1pp | 39.0% [38.7, 39.3] |
> | GPT-oss-20b | 78.1% [77.9, 78.3] | 67.0% [66.8, 67.3] | +11.1pp | 35.6% [35.3, 35.9] |
> | Gemma-3-27B | 79.5% [79.3, 79.7] | 67.9% [67.7, 68.2] | +11.6pp | 30.4% [30.1, 30.7] |
> | GPT-5-mini | 83.2% [82.6, 83.7] | 77.9% [77.4, 78.5] | +5.2pp | 21.0% [20.4, 21.7] |
> | Gemini-3-Flash | 84.2% [83.7, 84.7] | 79.4% [78.9, 80.0] | +4.8pp | 17.2% [16.6, 17.8] |
>
> Though the sample size is small, the stronger predictors (GPT-5-mini and Gemini-3-Flash) have higher baseline accuracy *and* higher accuracy with explanation. This means the *absolute gain* is smaller and is consistent with the reviewer’s intuition that stronger predictors will extract less *marginal* information since baselines are higher. However, the effect on NSG depends on the rate at which these components change. Furthermore, we expect the effect of stronger predictors to plateau fairly quickly. Indeed, Kendall's W = 0.75 rank agreement across predictors (Section 4.1) confirms that NSG rankings are robust to predictor strength, despite differences in absolute gain.
>
> We also note that the false-change rate (the percentage of time when the predator changes its answer conditioned on the fact that the reference model does not) shows that stronger predictors are less vulnerable to systematic bias, hence it should give a *tighter* bound on true explanation faithfulness by reducing the predictor-attributable noise floor. Our recommendation is to use the strongest predictors that are affordable.
>
> ### **Weakness 3: Domain scope**
>
> This is a fair critique and we will tighten the framing in the camera-ready version to be more specific. We will adjust the abstract and introduction to make this scope explicit, and reframe the safety discussion as motivational rather than as a direct claim supported by our evidence.
>
> ### **Weakness 4: "Technical concepts" labeling**
>
> Thank you—we agree this wording is imprecise. The distinction we intended is between *non-protected socioeconomic attributes* (Education, Occupation) and *protected demographic characteristics* (Race, Sex, Age), not "technical vs. non-technical." We will relabel accordingly in the camera-ready. As the reviewer notes, the underlying finding and discussion in Figure 7 is about articulating that unfaithfulness is driven by difficulty articulating complexity feature interactions rather than by incentives to conceal reliance on sensitive attributes for those datasets.
>
> ---
>
> ### **Question 1: Dimension-specific NSG for Moral Machines**
>
> This is a very useful suggestion. We will compute dimension-specific NSG for the Moral Machines domain and include it in the camera-ready appendix.
>
> ### **Question 3: Where does self-knowledge help most?**
> There has been a lot of interest in this result and we plan on adding more discussion to Appendix A.9. to include plots showing the points below.
> - **Datasets.** The self-knowledge advantage is most significant in the Breast Cancer (+4.6pp), Heart disease (+4.0pp), and Income (+4.0pp) datasets, and lowest in moral machines (+0.9pp). This is expected since self-model advantage should be lowest in the dataset where faithfulness is lowest.
> - **Model scale.** Models that have higher NSG tend to have higher self-knowledge advantage (r=0.66), which is expected, but the relationship between model scale and faithfulness is unclear in general. Interestingly, recent work published this month also found a complex relationship with scale, suggesting bigger models aren’t necessarily more faithful.
>
> ---
> Thank you again for your review! We appreciate the positive and actionable feedback. We agree with your recommendations and will implement them in the final version. Since this will strengthen the paper, we hope that you would consider increasing the score in line with our responses and edits.

---

> > ### Author Rebuttal · Reviewer_uVtH · 2026-04-04
> >
> > Thanks to the authors for the detailed responses. The new analysis on predictor strength (Weakness 2) is particularly useful, as the false-change rate metric is a nice addition that directly addresses the concern about disentangling predictor noise from explanation unfaithfulness. The commitments to tighten the framing around domain scope (Weakness 3), clarify the oracle ceiling interpretation (Weakness 1), and relabel 'technical concepts' (Weakness 4) are all appreciated. I will keep my score that reflects a positive assessment, which the rebuttal also confirms.

---

> > > ### Author Response · Authors · 2026-04-05
> > >
> > > Thank you for your detailed and constructive review, and for confirming that our responses address your concerns. We appreciate the time you’ve invested in helping us strengthen our paper.

---

### Official Review · Reviewer_52UZ · 2026-03-16

**Soundness:** 2
**Presentation:** 3
**Significance:** 3
**Originality:** 2
**Overall Recommendation:** 4
**Confidence:** 4

**Summary:**

This paper introduces Normalized Simulatability Gain (NSG) as a metric to measure the predictive information contained in an explanation. The underlying intuition is that faithful explanations should reveal the decision-making criteria of the model. The authors propose an evaluation protocol in which the model under evaluation (the reference model) first generates a pair *(answerₓ, explanationₓ)* for a given input *x*. A separate predictor model then uses either *(answerₓ)* alone or *(answerₓ + explanationₓ)* to simulate how the reference model would respond to related counterfactual inputs *C(x)*, where *C(x)* is constructed using the closest instances. The prediction accuracy with and without explanations is then used to compute the NSG. The method is evaluated on 18 large language models across three domains, and the results show that self-explanations improve the ability to predict the model’s behavior. They also demonstrate that self-explanation are not derivable from model input-output behaviour.

**Compliance With Llm Reviewing Policy:**

Affirmed.

**Final Justification:**

The rebuttal response have addressed most of my concerns. I encourage the authors to also discuss impact of optimization method used for instruction-tuned models.

**Key Questions For Authors:**

1. How sensitive are the evaluation to the template used?
2. How does the self-explanation improvement compare to model trained directly on the tabular data or tabular foundation models?

**Strengths And Weaknesses:**

* (+) The paper is clearly written and well structured.
* (+) The evaluation protocol is clearly defined and appears easily reproducible.
* (-) The main concerns relate to the robustness of the proposed method. The natural language conversion appears relatively straightforward, and the paper does not include any prompt sensitivity analysis, performance difference between base and instruct model, impact of post-training.
* (-) The paper does not sufficiently engage with the existing literature (e.g., the miscalibration of LLMs when used as risk scores (Cruz et al., 2024), or comparisons with tabular foundation models)

---

> ### Author Rebuttal · Authors · 2026-03-29
>
> Thank you for your helpful review! We appreciate you highlighting these points, and we think addressing these will significantly strengthen the manuscript.
>
> ---
>
> ### **Prompt sensitivity**
>
> We agree with your recommendation to add prompt sensitivity analysis to the paper to demonstrate robustness. Running this across the full evaluation is computationally prohibitive, but we’ve run a targeted experiment with a single reference model (Qwen3 32B), and will include this in the appendix.
>
> We considered three variations of our original prompt templates on three datasets (Heart Disease, Income and Breast Cancer Recurrence). We observe no significant impact on NSG using semantically similar, but different prompts:
>
> | Variant | NSG (%) | 95% CI | Acc w/ | Acc w/o |
> | :--- | :--- | :--- | :--- | :--- |
> | V1 (baseline) | 27.4 | [23.7, 30.8] | 80.5 | 73.2 |
> | V2 (professional) | 29.9 | [26.4, 33.4] | 79.9 | 71.4 |
> | V3 (specialist) | 26.3 | [22.8, 29.7] | 78.9 | 71.3 |
>
> This experiment also points to an interesting direction for future work. Prompt optimization methods such as GEPA (https://arxiv.org/pdf/2507.19457) have proved effective at increasing performance in various tasks. Finding prompts that produce more faithful explanations would be of great interest to the community. We will highlight this as a useful future direction in the paper.
>
> ### **Risk calibration discussion**
>
> We think it would be valuable to include a discussion of this in the related work, and will do so in the camera ready version. Whether models can accurately verbalize their internal risk scores (Cruz et al.) is a closely related question to whether they can faithfully verbalize a natural language explanation.
>
> As it happens, our original experiments already elicited verbalized confidence scores (as “high” / “medium” / “low”) so we can run analysis post-hoc and add a thorough section to the appendix. We will discuss a few initial points here:
>
> * *Average simulatability gain is positive at all confidence levels:* Measuring NSG across predictors and models (399,126 counterfactual prediction pairs), simulatability gain is statistically positive at each confidence level.
> * *Correlation with verbalized confidence:* In some model families (notably, *all* proprietary models: Gemini 3, GPT-5, Claude 4.5) we see a positive correlation with average simulatability gain and verbalized confidence. This is particularly notable in the Claude models (see table). This is a noteworthy result and we thank the reviewer for suggesting this.
> | Reference Model | Sim. Gain (High confidence)  | Sim. Gain (Medium confidence) | Sim. Gain (Low confidence)|
> |---|---|---|---|
> | Claude-Opus-4.5 | +0.168 [+0.155, +0.180] | +0.089 [+0.081, +0.096] | +0.029 [+0.000, +0.060] |
> | Claude-Sonnet-4.5 | +0.152 [+0.142, +0.162] | +0.111 [+0.103, +0.120] | +0.041 [-0.012, +0.094] |
> | Claude-Haiku-4.5 | +0.127 [+0.117, +0.138] | +0.097 [+0.089, +0.104] | +0.018 [-0.026, +0.061] |
> * *Calibration:* Consistent with Cruz et al., we find that verbalized confidence is poorly calibrated with respect to ground truth accuracy in our datasets (though our resolution is limited with use of a coarse three-level scale for confidence). This is interesting, as the above correlation appears to hold despite this lack of calibration, i.e., for some models, verbalised confidence scores appear to be predictors of explanatory faithfulness, even when they are not calibrated.
>
> ### **Tabular foundation models**
>
> Our motivations for this paper were to measure the properties of frontier LLMs, to enable better assessments of explanation-related risks. To our knowledge, current tabular foundation models (e.g., TabPFN) are not designed to produce the natural language explanations our metric is designed to evaluate. More broadly, we evaluate LLMs because these are the models being deployed in high-stakes settings where explanation faithfulness matters most and users may rely on self-explanations to assess a model’s decision-making. Importantly, our analysis decouples ground truth accuracy from faithfulness, as NSG does not depend on ground truth labels.
>
> ### **Comparing base/instruct models**
>
> Prior work has examined this question directly. Appendix C of https://arxiv.org/abs/2503.13445 analyzes differences in faithfulness between base and instruction-tuned models. Conceptually, base and instruct models differ in a way that makes direct comparison non-trivial: since base models are trained to predict the most likely text completion, the most standard way of getting them to give an “explanation” which we are aware of is via few-shot prompting (https://arxiv.org/abs/2005.14165). However, this tests the model’s ability to replicate the quality of the demonstration explanations, rather than independently generate faithful explanations, which is less aligned with the overall goal of this work.
>
> ---
>
> We hope these discussions address your concerns and you will consider raising your score.

---

> > ### Author Rebuttal · Reviewer_52UZ · 2026-04-04
> >
> > Thanks for the rebuttal responses.
> > Most of my concerns have been addressed, and I will increase my score.
> >
> > Short note on the impact of post-training. Even when restricting to instruction-tuned models, the choice of optimization method (e.g., SFT, DPO, RLHF) can significantly affect the nature of the explanations. It would be interesting to discuss this aspect as well.

---

> > > ### Author Response · Authors · 2026-04-05
> > >
> > > Dear Reviewer 52UZ,
> > >
> > > Thank you for your reply and for updating your score. We are very pleased with the additional experiments and insights that emerged from your suggestions; they will greatly improve the paper.
> > >
> > > **Post-training**: Understanding the influence of post-training on faithfulness is an important open question. The choice of alignment method (SFT, DPO, RLHF) may meaningfully influence how models articulate explanations, particularly around sensitive attributes. Existing work has also briefly touched on why this might be the case theoretically [1]. Our feature-level unfaithfulness analysis (Figure 7) provides a natural starting point for examining this. We plan to add a discussion section exploring whether models with heavier safety-oriented post-training show different patterns of unfaithfulness on protected vs. non-protected features.
> > >
> > > *Proposed additional experiment*: In the camera-ready version we can include some experiments in this direction. The Olmo 3 models include public post-training intermediate checkpoints. We can measure how faithfulness is affected by the different training stages. A more detailed project would be valuable follow-up work, and we will promote this in the paper.
> > >
> > > We recognize that the scores for this manuscript may place it in borderline territory. If there is anything major still holding this back from a clear accept, we would welcome the feedback and commit to addressing it in the camera-ready.
> > >
> > > Thank you again for putting time into helping improve our paper.
> > >
> > > ---
> > >
> > > [1] Are DeepSeek R1 And Other Reasoning Models More Faithful? Chua and Evans, 2025.

---

### Decision · Program_Chairs · 2026-04-30

**Decision:**

Accept (regular)

**Comment:**

Summary. This paper introduces the Normalized Simulatability Metric (NSG), a new metric for measuring the faithfulness of LLM explanations by quantifying how much an explanation helps an observer predict the model's behavior on related inputs. Extensive evaluations show that self-explanations improve prediction of model behavior (11-37% NSG) and outperform external explanations from stronger models, suggesting a self-knowledge advantage. The metric also reveals that 5-15% of self-explanations can be egregiously misleading, offering a nuanced view of the current state of LLM explanations.

Reviewers' consensus. 3 reviewers suggest acceptance (2 accept, 1 weak accept), 1 reviewer suggest rejection (1 weak reject). The main criticism relate to the robustness of the proposed approach, scope of literature cited, and novelty.

Assessment. The paper proposes a faithfulness metric that closes the evolution gap in frontier LLMs. The authors responded thoroughly to the reviewers, and the results are extensive and support the conclusion.